# Silicon Fractionation of Soluble Silicon in Volcanic Ash Soils That May Affect Groundwater Silicon Content on Jeju Island, Korea

**Won-Pyo Park [1]** [ID]**, Hae-Nam Hyun [1] and Bon-Jun Koo [2],***

[1]  Major of Plant Resources and Environment, Jeju National University, Jeju 63243, Korea;
   oneticket@jejunu.ac.kr (W.-P.P.); hnhyun@jejunu.ac.kr (H.-N.H.)
[2]  Department of Biological Sciences, California Baptist University, Riverside, CA 92504-3297, USA
*   Correspondence: bonjunkoo@calbaptist.edu; Tel.: +1-951-343-4621

**Abstract:** Silicon (Si) is found in various fractions of soil, depending on the pedogenic processes of the environment. Dissolved Si (DSi) is adsorbed in soil particles or leaches through the soil profile into the groundwater. The objective of this study is to quantify, using the sequential extraction method, the different Si fractions in volcanic ash soils on Jeju Island that may affect groundwater Si content, and to compare them with those in forest soils on mainland Korea. Most of the Si in these soils was bound in unavailable forms as primary and secondary silicates. The second largest proportion of Si in the non-Andisols of Jeju Island and Korean mainland soils was accumulated as amorphous Si, while in the Andisols of Jeju Island, the second most significant Si fraction was in pedogenic oxides and hydroxides. The products of these soil formations were short-range-order minerals such as allophane (4–40%). The adsorbed Si concentration tended to increase at lower depths in Andisols ($100–1400 \ mg \ kg^{-1}$) and was approximately five times higher than that in non-Andisols. The results indicate that Si is more soluble in the Andisols of high precipitation regions and that Andisols on Jeju Island potentially affect groundwater Si concentration.

**Keywords:** allophane; Andisol; readily soluble silicon; sequential silicon extraction

## 1. Introduction

On oceanic islands, groundwater aquifers replenished by infiltrating rainfall are a major source of freshwater. About 92% of the drinking water on Jeju Island, the largest volcanic island in Korea, is supplied from groundwater [1]. In South Korea, bottled water is very popular and is produced from groundwater sources on Jeju Island [2]. As the consumption of bottled water increases, interest in the quality of the drinking water also grows. A high concentration of silicon (Si) along with Ca and K in drinking water improves water taste and benefits human health [3]. Many researchers have reported that Si in drinking water inhibits the absorption of Al in the body, which is effective in preventing Alzheimer's disease and has effects on atherosclerosis, bone formation, and cholesterol levels [4–8].

Si dissolves into mono-silicic acid ($H_4SiO_4$) by the weathering of silicate minerals [9]. Dissolved Si (DSi) in the soil solution is absorbed by plants, and leached Si is moved from the river to the sea. Jeju Island has no permeant stream or river due to the high permeability of its volcanic ash soils and lava and pyroclastic flows [10,11]. Hence, much of the precipitation permeates the soil and enters groundwater, which makes it easy to leach DSi into groundwater. The median Si concentration in Jeju Island's groundwater is 33.6 mg $SiO_2 \ L^{-1}$ (up to 63 mg $SiO_2 \ L^{-1}$), which is 1.8 times higher than that in bottled water from mainland Korea [12,13]. Si accounts for more than 80% of the ingredients that make Jeju Island's groundwater taste good [12]. Previous studies on the

hydro-chemical composition of Jeju Island's groundwater were mostly conducted for preservation and control against contamination. In particular, their major focus included increasing Na and Cl concentrations by mixing with seawater, nitrogenous fertilizers used on farmland, and nitrate nitrogen originating from livestock wastewater [14,15].

Since it is easy on Jeju Island for DSi to move below the soil and flow into groundwater, the Si concentration in groundwater has a close correlation with the solubility of Si compounds in the soil. Si in the soil consists of primary crystalline silicates (e.g., quartz, feldspars), secondary silicates (e.g., clay minerals), poorly crystalline or microcrystalline silicates (e.g., allophane, imogolite, opal-CT), and amorphous silica (biogenic and pedogenic silica) [9,16].

Si released by weathering is either dissolved into the soil solution or absorbed into soil particles and exists in various forms, such as mono-silicic acid, poly-silicic acid, or organic-inorganic complexes [9,16,17]. As pyroclastic materials in volcanic ash soil are weathered quickly, Si, Al, and Fe are dissolved easily, and because Al preferentially binds with organic matter and produces Al-humus complexes in warm, humid climate conditions, a large amount of DSi can exist in the soil solution [18,19]. This DSi concentration decreases under high precipitation conditions, and DSi exists as monomeric Si or binds with polymerized Al to produce allophane [19,20]. If allophane undergoes extreme desilification over time, it sometimes produces gibbsite [19,21,22]. Eighty percent of Jeju Island's soils are Andisols, which consist of allophane and Al-humus complexes and are mainly distributed in the island's center and southeast regions where precipitation is high. In the coastal and middle mountainous areas of Jeju Island's west and north where precipitation is relatively low and evapotranspiration is high, non-Andisol soils (e.g., Alfisols, Inceptisols, Mollisols, and Ultisols) mainly consist of layered silicates [23,24]. DSi that fails to form layered silicates and allophane during weathering is likely to exist either as weakly bonded forms within the soil or in groundwater due to downward movement by rainwater.

Georgiadis et al. [25] reported that water-soluble and adsorbed Si, among other Si fractions from Si sequential extraction, is easy to mobilize. In addition, amorphous Si and Si occluded in pedogenic oxides and hydroxides can become potential sources of easy-to-mobilize Si because amorphous Si is dissolved more easily than crystalline minerals [26]. Miretzky et al. [27] reported that the dissolution of amorphous Si from volcanic pyroclastic or biogenic origin most likely determines the high DSi concentration in groundwater. Park et al. [23] compared the soluble Si content in soils from Jeju Island and mainland Korea to identify why Jeju Island's groundwater has a higher DSi content. However, since they used the 1N sodium acetate extraction method, the soluble Si content in the analyzed soils included Si in silicate minerals, amorphous silica, and allophane. Thus, it is necessary to estimate the readily dissolvable Si content by quantifying different Si fractions from compounds in the soils. Since precipitation and permeability are high on Jeju Island, Si can be easily dissolved in the soil solution and then continue to be leached, which may contribute to increasing the $SiO_2$ concentration in groundwater.

This study compared the $SiO_2$ concentration in groundwater from Jeju Island and mainland Korea. We compared the distribution characteristics of sequentially extracted Si forms in Andisols (Noro and Pyeongdae series), Ultisols (Jeju series), and Alfisols (Gangjeong series) on Jeju Island, the parent materials of which are pyroclastics, and Inceptisols (Samgag and Oesan series) derived from granite and granite gneiss on mainland Korea. This study will illustrate a correlation between the content of soluble Si in the soil and the DSi concentration in groundwater.

## 2. Materials and Methods

### 2.1. Sampling and Analyzing Groundwater

Sixty-five groundwater wells were selected for Jeju Island's groundwater samples. After opening the well valve and letting water flow for about 5 min, the groundwater samples were collected into polyethylene bottles (500 mL). For the samples, 2 mL of concentrated nitric acid solution at 68% was added

into the samples to reach a 0.5% nitric acid concentration. The samples were transported to the laboratory, filtered by a 0.45 μm glass fiber filter, measured for Si concentration by inductively coupled plasma optical emission spectrometry (ICP-OES) (Integra XL dual GBC, Melbourne, Australia), and calculated as $SiO_2$. Data analyzed in Gangwon Province (58 samples), Gyeonggi Province (64 samples), Chungcheong Province (88 samples), Gyeongsang Province (126 samples), and Jeolla Province (103 samples) were provided by Geochemistry Laboratory, Korea University, and used for the groundwater $SiO_2$ concentrations in mainland Korea in comparison to those of Jeju Island's groundwater.

### 2.2. Soil Samples

We analyzed 28 soil samples collected from horizons with six different soil profiles in Jeju Island and mainland Korea (Figure 1). For the volcanic ash soil samples from Jeju Island, four soils with different properties (Noro, Pyeongdae, Jeju, and Gangjeong series) were selected according to the Taxonomical Classification of Korean Soils [28]. To compare with volcanic ash soils on Jeju Island, two soils (Samgag and Oesan series) from mainland Korea were selected, which were representative soils (typifying Pedon) collected for reclassification research on Korean soils [28] (Table 1). The parents of the four soil series on Jeju Island are pyroclastic materials. The Noro and Pyeongdae series are classified as Andisols. While the Noro series is a cinder cone soil distributed in high elevation mountainous areas (646 m asl) south of Hallasan on Jeju Island, the Pyeongdae series is distributed in lava plains (320 m asl) east of Jeju Island. The Jeju and Gangjeong series are classified as Ultisols and Alfisols, respectively, and have argillic horizons, or layers of accumulated clay. These two soils are distributed in 2–7% sloped lava plains north of Jeju Island. The Samgag and Oesan series on mainland Korea are developed from residuum parent materials in granite and gneiss, respectively, located in 30–60% sloped mountainous areas, and are classified as Inceptisols.

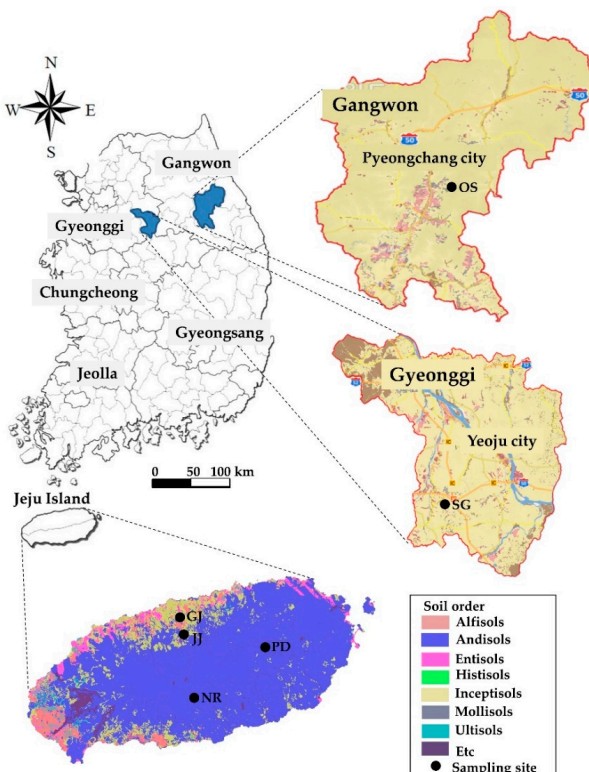

**Figure 1.** Map of the study area with soil sampling locations in Jeju Island [Andisols: Noro (NR) and Pyeongdae (PD), Ultisols: Jeju (JJ), Alfisols: Gangjeong (GJ)] and in Korean mainland forest soils [Inceptisols: Samgag (SG) and Oesan (OS)].



**Table 1.** Site characteristics of six soil profiles.

| Region | Soil Series | MAP [a] | Land Use/Vegetation | Parent Material | Coordinates | Soil Taxonomy/WRB [b] |
|---|---|---|---|---|---|---|
| Jeju | Noro (NR) | 3430 | Forest/Latifoliate trees | Pyroclastic materials | 33°18′17.4″ N 126°28′19.1″ E | Acrudoxic Fulvudands/Fulvic Silandic Andosols |
| Jeju | Pyeongdae (PD) | 2660 | Pasture/Wide grasses | Pyroclastic materials | 33°26′01.5″ N 126°43′30.4″ E | Acrudoxic Melanudands/Melanic Silandic Andosols |
| Jeju | Jeju (JJ) | 1860 | Forest/Pinus thunbergii, | Pyroclastic materials | 33°26′58.5″ N 126°31′09.7″ E | Andic Palehumults/Umbric Cutanic Alisols |
| Jeju | Gangjeong (GJ) | 1740 | Forest/Pinus thunbergii, | Pyroclastic materials | 33°27′36.8″ N 126°29′07.7″ E | Mollic Paleudalfs/ Cutanic Luvisols |
| Gyeonggi | Samgag (SG) | 1370 | Forest/Pinus | Granite | 37°11′17.9″ N 127°36′36.5″ E | Typic Dystrudepts/Haplic Cambisols |
| Gangwon | Oesan (OS) | 1220 | Forest/Coniferous and deciduous | Mica schist, mica gneiss | 37°28′8.4″ N 128°33′9.6″ E | Typic Dystrudepts/ Leptic Cambisols |

[a] Mean annual precipitation (mm year$^{-1}$); [b] Taxonomical Classification of Korean Soils [28].

## 2.2.1. Physicochemical Analyses of Soils

The soil samples were air-dried and passed through a 2-mm sieve for analysis (Table 2). Unless otherwise noted, the soils' physicochemical properties were analyzed according to the Soil Survey Laboratory Methods Manual [29]. Bulk density (Bd) was determined by the dry weight of 100 cm$^3$ undisturbed core samples. After removing organic matter, the sand, silt, and clay contents were measured by wet sieving (sand fractions) and pipette (silt and clay) methods. Soil pH was measured by the Orion Star A211 pH meter (Thermo Fisher Scientific, Waltham, MA, USA) at a 1:2 ratio of soil to 0.01 M CaCl$_2$ solution. Soil organic carbon (SOC) was analyzed by the Walkley and Black wet digestion method [30].

**Table 2.** Physical and chemical properties of the soils studied in Jeju Island (Noro, Pyeongdae, Jeju, and Gangjeong) and mainland Korea (Samgag and Oesan).

| Horizon | Depth (cm) | Bd (g cm$^{-3}$) | pH (CaCl$_2$) | SOC (g kg$^{-1}$) | Sand (%) | Site | Clay | Fe$_d$ | Al$_o$ + 1/2Fe$_o$ | Allo. | Si/Al (Molar Ratio) |
|---|---|---|---|---|---|---|---|---|---|---|---|
| Noro | | | | | | | | | | | |
| A | 0–22 | 0.49 | 5.39 | 122 | 12.1 | 54.8 | 33.0 | 2.59 | 6.10 | 11.6 | 2.12 |
| BA | 22–44 | 0.53 | 5.32 | 70.1 | 30.0 | 64.8 | 5.19 | 3.65 | 7.64 | 20.9 | 1.76 |
| Bw | 44–87 | 0.50 | 5.76 | 31.2 | 28.7 | 62.0 | 9.34 | 3.82 | 10.3 | 30.8 | 1.61 |
| C | 87–160 | 0.51 | 5.90 | 13.9 | 48.7 | 43.2 | 8.17 | 1.88 | 11.8 | 39.0 | 1.51 |
| Pyeongdae | | | | | | | | | | | |
| A | 0–18 | 0.51 | 5.24 | 120 | 26.3 | 38.8 | 34.8 | 4.59 | 5.44 | 7.27 | 2.19 |
| AB | 18–44 | 0.52 | 5.19 | 87.3 | 13.3 | 65.5 | 21.2 | 4.50 | 7.47 | 15.9 | 2.41 |
| Bw1 | 44–73 | 0.60 | 5.35 | 43.7 | 15.5 | 63.1 | 21.4 | 4.08 | 6.17 | 13.3 | 2.44 |
| Bw2 | 73–100 | 0.71 | 5.46 | 23.0 | 10.1 | 61.9 | 28.0 | 4.13 | 7.53 | 15.2 | 2.12 |
| BC | 100–160 | 0.72 | 5.74 | 15.4 | 34.3 | 47.1 | 18.6 | 3.92 | 8.34 | 19.5 | 2.01 |
| Jeju | | | | | | | | | | | |
| Ap | 0–20 | 0.92 | 5.08 | 37.8 | 7.71 | 58.6 | 33.7 | 1.92 | 2.46 | - | 5.69 |
| AB | 20–41 | 1.03 | 5.02 | 19.9 | 3.29 | 59.9 | 36.8 | 2.10 | 2.13 | - | 6.23 |
| Bt1 | 41–65 | 1.29 | 5.13 | 9.80 | 2.65 | 62.4 | 35.0 | 2.23 | 1.45 | - | 6.16 |
| Bt2 | 65–92 | 1.48 | 5.19 | 7.12 | 3.97 | 61.0 | 35.1 | 2.42 | 1.26 | - | 6.07 |
| Bt3 | 92–160 | 1.48 | 5.12 | 6.40 | 5.06 | 57.6 | 37.3 | 2.54 | 1.21 | - | 6.21 |
| Gangjeong | | | | | | | | | | | |
| Ap | 0–24 | 1.05 | 5.52 | 30.2 | 10.6 | 66.1 | 23.3 | 1.82 | 1.38 | - | 6.36 |
| BAt | 24–38 | 1.36 | 5.22 | 13.9 | 4.76 | 70.1 | 25.1 | 1.94 | 0.74 | - | 6.35 |
| Bt1 | 38–53 | 1.48 | 5.29 | 5.15 | 5.09 | 63.5 | 31.4 | 2.12 | 0.59 | - | 6.75 |
| Bt2 | 53–85 | 1.53 | 5.27 | 2.86 | 3.82 | 59.9 | 36.3 | 2.32 | 0.59 | - | 6.78 |
| Bt3 | 85–160 | 1.53 | 5.23 | 1.92 | 3.87 | 58.2 | 38.0 | 1.93 | 0.89 | - | 6.74 |

**Table 2.** *Cont.*

| Horizon | Depth (cm) | Bd (g cm$^{-3}$) | pH (CaCl$_2$) | SOC (g kg$^{-1}$) | Sand (%) | Site | Clay | Fe$_d$ | Al$_o$ + 1/2Fe$_o$ | Allo. | Si/Al (Molar Ratio) |
|---|---|---|---|---|---|---|---|---|---|---|---|
| Samgag | | | | | | | | | | | |
| A | 0–15 | 1.05 | 4.27 | 22.8 | 78.4 | 13. | 7.81 | 0.24 | 0.11 | - | 4.25 |
| BA | 15–32 | 1.36 | 4.64 | 9.44 | 74.9 | 16.5 | 8.58 | 0.25 | 0.11 | - | 4.22 |
| Bw | 32–50 | 1.48 | 4.77 | 7.69 | 74.0 | 16.6 | 9.37 | 0.32 | 0.11 | - | 4.05 |
| C1 | 50–78 | 1.53 | 5.19 | 3.94 | 70.9 | 15.2 | 13.9 | 0.45 | 0.08 | - | 4.03 |
| C2 | 78–180 | 1.53 | 5.43 | 2.39 | 69.7 | 14.9 | 15.4 | 0.51 | 0.11 | - | 3.53 |
| Oesan | | | | | | | | | | | |
| A | 0–15 | - | 4.77 | 61.1 | 23.8 | 49.6 | 26.6 | 1.40 | 1.15 | - | 5.72 |
| BA | 15–34 | - | 4.90 | 24.3 | 21.0 | 54.4 | 24.6 | 1.28 | 1.09 | - | 6.41 |
| Bw | 34–63 | - | 4.95 | 11.6 | 24.1 | 54.1 | 21.8 | 1.14 | 0.87 | - | 6.31 |
| C | 63–98 | - | 5.22 | 5.13 | 48.3 | 38.7 | 13.0 | 0.87 | 0.62 | - | 6.92 |

Bd, bulk density; SOC, soil organic carbon; Fe$_d$, dithionite-citrate-extractable Fe; Fe$_o$, Al$_o$, acid-oxalate-extractable Al$_o$ and Fe$_o$; Allo., Allophane; -, not determined.

Organically complexed Al (Al$_p$) was extracted for 16 hours using 0.1 M sodium pyrophosphate (pH 10). Free iron oxides (Fe$_d$), which consist of crystalline and non-crystalline hydrous oxides, were extracted for 16 h using 0.4 g sodium dithionite and 0.57 M sodium citrate. Al, Fe, and Si (Al$_o$, Fe$_o$, Si$_o$, respectively) included in organically complexed Al, non-crystalline hydrous oxides of Fe and Al, and allophane were protected from light and extracted for 12 hours in the dark with 0.2 M acid ammonium oxalate (pH 3.0) using a mechanical vacuum extractor (SampleTek, MAVCO, Lawrenceburg, KY, USA). The allophane content was calculated by the equation $100 \times Si_o/\{-5.1[(Al_o - Al_p)/Si_o$ atomic ratio] + 23.4\} [31]. The total Si and Al in the soils were dissolved in HNO$_3$, HCl, HF, and H$_3$BO$_3$ in a microwave oven. The Al, Fe, and Si contents were measured by ICP-OES (JY 138 Ultrace, Jobin Yvon, Longjumeau, France).

X-ray diffraction (XRD) analyses were performed by a Geigerfiex 2301 diffractometer (Rigaku, Tokyo, Japan), using Cu Kα radiation at 30 kV and 15 mA to measure powder samples and clay fractions of less than 2 μm treated by heating and ethylene glycol at 550 °C. Semi-quantitative analysis (Table S1) was performed using a quantitative analysis program (SIEROQUANT™, version 3.0).

2.2.2. Sequential Si Extraction Procedures

The sequential Si extraction method developed by Georgiadis et al. [32] was used for different Si fractions in the soils, in combination with other previous methods (Table 3).

**Table 3.** Modified sequential extraction procedure for Si fractions in soil.

| Step | Si Fraction | Extractant | Extraction Conditions |
|---|---|---|---|
| 1 | Mobile Si | 0.01 M CaCl$_2$ | SSR [a] 1:10 at 20–25 °C, 24 h |
| 2 | Adsorbed Si | 0.01 M acetic acid | SSR 1:10 at 20–25 °C, 24 h |
| 3 | Si bound to soil organic matter | Hot concentrated H$_2$O$_2$ | SSR 1:30 in a water bath at 85 °C until the reaction is completed |
| 4 | Si occluded in pedogenic oxides and hydroxides | 0.2 M Ammonium oxalate (pH 3) | SSR 1:50 at 20–25 °C in the dark, 4 h |
| 5 | Si in total amorphous silica | 0.2 M NaOH | SSR 1:400 at 20–25 °C in the steps (5 to 240 h) |
| 6 | Residual Si in crystalline silicates [b] | | |
| | Total Si [c] | | |

[a] SSR = soil to solution ratio. [b] Residual Si = Total Si minus the sum of the Si fractions extracted in Steps 1 to 5 above. [c] Total Si = digestion with a mixture of HNO$_3$, HCl, HF, and H$_3$BO$_3$.

Hence, to extract Si in allophane produced from volcanic ash soils, this study used 0.2 M ammonium oxalate extraction in darkness [31,33–35]. As amorphous silica can be extracted as biogenic and minerogenic Si [32], both reservoirs are potential sources for mobile Si [25]. Accordingly, this study did not distinguish between the two forms of Si but extracted Si in total amorphous silica.

One gram of soil sample was placed in a 50-mL centrifuge tube with a polyethylene screw cap and extracted sequentially with the extractant for each step. Mobile Si was extracted by adding 10 mL 0.01 M $CaCl_2$ solution and shaking it at 100 rpm for 24 h. Adsorbed Si was extracted by adding 10 mL 0.01 M acetic acid solution and shaking it at 100 rpm for 24 h. After adding 20 mL 17.5% $H_2O_2$ solution, shaking it 4–5 times, leaving it for 1 hour at 20–25 °C adding 10 mL 35% $H_2O_2$ solution, and placing the sample in a water bath at 85 °C, Si bound to soil organic matter was reacted to completion. After adding 50 mL 0.2 M ammonium oxalate solution (pH 3) and protecting it from light, Si occluded in pedogenic oxides and hydroxides was shaken for 4 h in tubes wrapped with aluminum foil. After adding 400 mL 0.2 M sodium hydroxide solution, Si in total amorphous silica was shaken slowly for 10 days at 20–25 °C at 100 rpm. After 5, 24, 48, 72, 120, 144, 168, 192, 216, and 240 h of shaking, 5 mL of the solution was drawn by pipette, filtered by a 0.45-μm syringe filter, diluted, and then analyzed (Figure S1). The amount of Si in total amorphous silica was determined by SigmaPlot 10.0 software (Systat Software Inc., Chicago, IL, USA) using the first-order dissolution model [36]. Residual Si in crystalline silicates was calculated as the difference between total Si and the sum of the Si fractions extracted in steps 1–5. All extractions were executed on four replicates. After each extraction step, the extracts were centrifuged at 3000 rpm for 15 min to separate soils and supernatants. To remove the extract from the previous step after separating the supernatant, 10 mL distilled water was added, and the solution was centrifuged in each extraction step. Si in the extract was analyzed by ICP-OES (JY 138 Ultrace, Jobin Yvon).

Additionally, for the Bw horizon in the Noro series, XRD and scanning electron microscope/ energy-dispersive X-ray spectroscopy (SEM/EDS) analyses were performed on the soils before sequential extraction and on residual soils after steps 4 and 5 to observe allophane and crystalline minerals in the soils. SEM/EDS analysis was performed at an acceleration voltage of 15 kV using FE-SEM SUPRA25 (Zeiss, Jena, Germany) and ELPHY Quantum (Raith, Dortmund, Germany).

## 2.3. Statistical Analyses

SPSS 18.0 (SPSS Inc., Chicago, IL, USA) was used for statistical analyses, and correlation analysis was evaluated by Spearman's rank correlation coefficients due to the partially normal and non-normal distribution patterns of the data.

## 3. Results and Discussion

### 3.1. Comparison of the Silica Concentration in Groundwater between Jeju Island and Mainland Korea

The $SiO_2$ concentration in groundwater from Jeju Island and mainland Korea is illustrated in Figure 2. The $SiO_2$ concentration in Jeju Island's groundwater ranged from 29.5 to 66.7 mg $L^{-1}$ (average of 39.2 mg $L^{-1}$). The $SiO_2$ concentration in mainland Korea's groundwater ranged from 0.79–29.2 mg $L^{-1}$ (average of 10.3 mg $L^{-1}$), and the average $SiO_2$ concentration in Gangwon Province's groundwater was the lowest at 6.0 mg $L^{-1}$. These results were consistent with those previously reported, showing that the $SiO_2$ concentration in groundwater derived from volcanic rocks on Jeju Island was higher than that from granite, gneiss, and metamorphic rocks [2,13]. The silica concentration in groundwater in Argentina, which is similarly of volcanic pyroclastic origin, ranges from 19.6–72.9 mg $L^{-1}$, similar to the silica content in Jeju Island's groundwater [27]. Weathering in quartz, amorphous silica, and aluminosilicate minerals may have an effect on the Si concentration in groundwater [2,9,16,27]. In general, extrusive rocks (e.g., basalt, andesite) have a lower total Si content but are weathered more quickly than intrusive rocks (e.g., granite) [27,37]. Therefore, weathering in Jeju Island's basaltic pyroclastic materials releases a higher percentage of soluble Si than do granitic rocks.

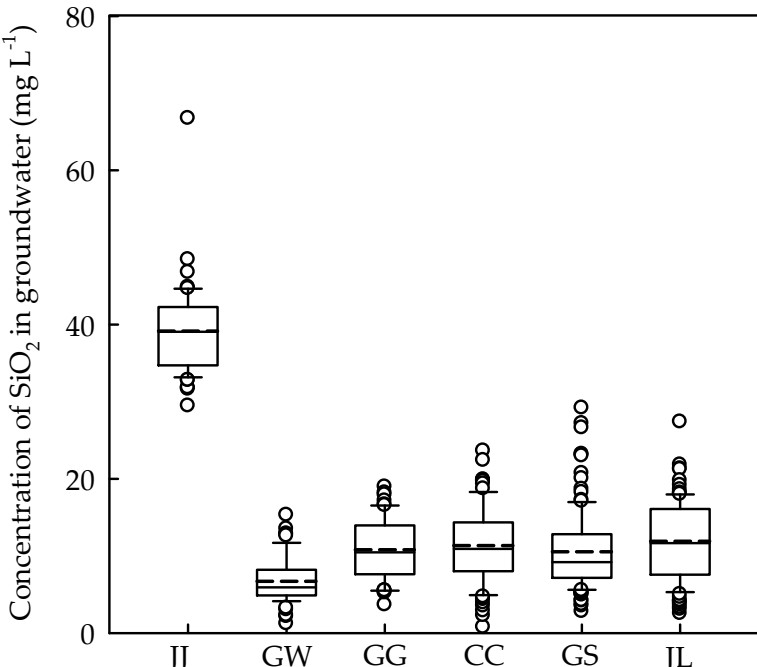

**Figure 2.** Box plots of SiO$_2$ concentration in groundwater of South Korea. JJ: Jeju Island, GW: Gangwon province, GG: Gyeonggi province, CC: Chungcheong province, GS: Gyeongsang province, JJ: Jeolla province. Boxes represent 25th, 50th (median), and 75th percentiles, and the whiskers indicate the minimum and maximum values. Mean values (dashed line); outliers (o).

As precipitation increases while pyroclastic materials are weathered, the leached amount of the base and Si increases while Al is preserved. Hence, the Si concentration in the soil solution in Andisols decreases, producing more noncrystalline materials (e.g., allophane, ferrihydrite, and noncrystalline hydrous oxides of Fe and Al) than crystalline layer silicates [18–20]. Under wet climate conditions, Si losses in volcanic ash soils have been reported by mass-balance calculations [38–40]. In this study's soils, the SiO$_2$/Al$_2$O$_3$ molar ratio was below 2 in Andisols, while other soils showed higher ratios from 3.5–7 (Table 3). The low SiO$_2$/Al$_2$O$_3$ molar ratio in Andisols suggests that a large amount of DSi in the soil solution has been leached during the pedogenic processes. A large amount of DSi that fails to form layered silicates or allophane during the soils' pedogenesis (depending on weathering and climate conditions) may exist as readily soluble Si in Si pools from Jeju Island's volcanic ash soils.

*3.2. Sequential Si Extraction Procedures*

3.2.1. Mobile and Adsorbed Si

In sequential extractions, we compared the Si fraction content among Andisols, Ultisols, and Alfisols from Jeju Island and Inceptisols from mainland Korea. The Si fraction content varied greatly depending on the depth and development of the soil (Figure 3, Table S2).

The mobile Si content in the soil profiles of Jeju Island and mainland Korea ranged from 6.9–57.1 µg·g$^{-1}$ and increased with increasing soil depth (Figure 3a). The highest mobile Si was recorded as 57.1 and 52.9 µg·g$^{-1}$ in the C horizon from the Noro series, Jeju Island's Andisols, and the Bt3 horizon from the Gangjeong series Alfisols. Compared with the mobile Si content, the adsorbed Si content increased more along with increasing soil depth in Andisols than in other soils. The adsorbed Si content was similar in topsoil horizons across the soils, ranging from 9.4–24 µg·g$^{-1}$. In subsoil horizons, however, the adsorbed Si content in Andisols increased up to 400 µg·g$^{-1}$, while it ranged from 16–35 µg·g$^{-1}$ in subsoil horizons from other soils, a slightly greater increase than in topsoil horizons (Figure 3b). In soils other than Andisols, the mobile and adsorbed Si contents were similar to the mobile Si (0.3–28 µg·g$^{-1}$) and adsorbed Si (1.2–39 µg·g$^{-1}$) contents in six soil profiles from Southwest

Germany [25]. Nevertheless, the adsorbed Si content in Andisols from Jeju Island was 20 times higher in subsoil horizons than in other soils. This seems attributable to the fact that DSi may have been leached from topsoil horizons to subsoil horizons since the areas in Jeju Island where Andisols are developed have a high level of precipitation.

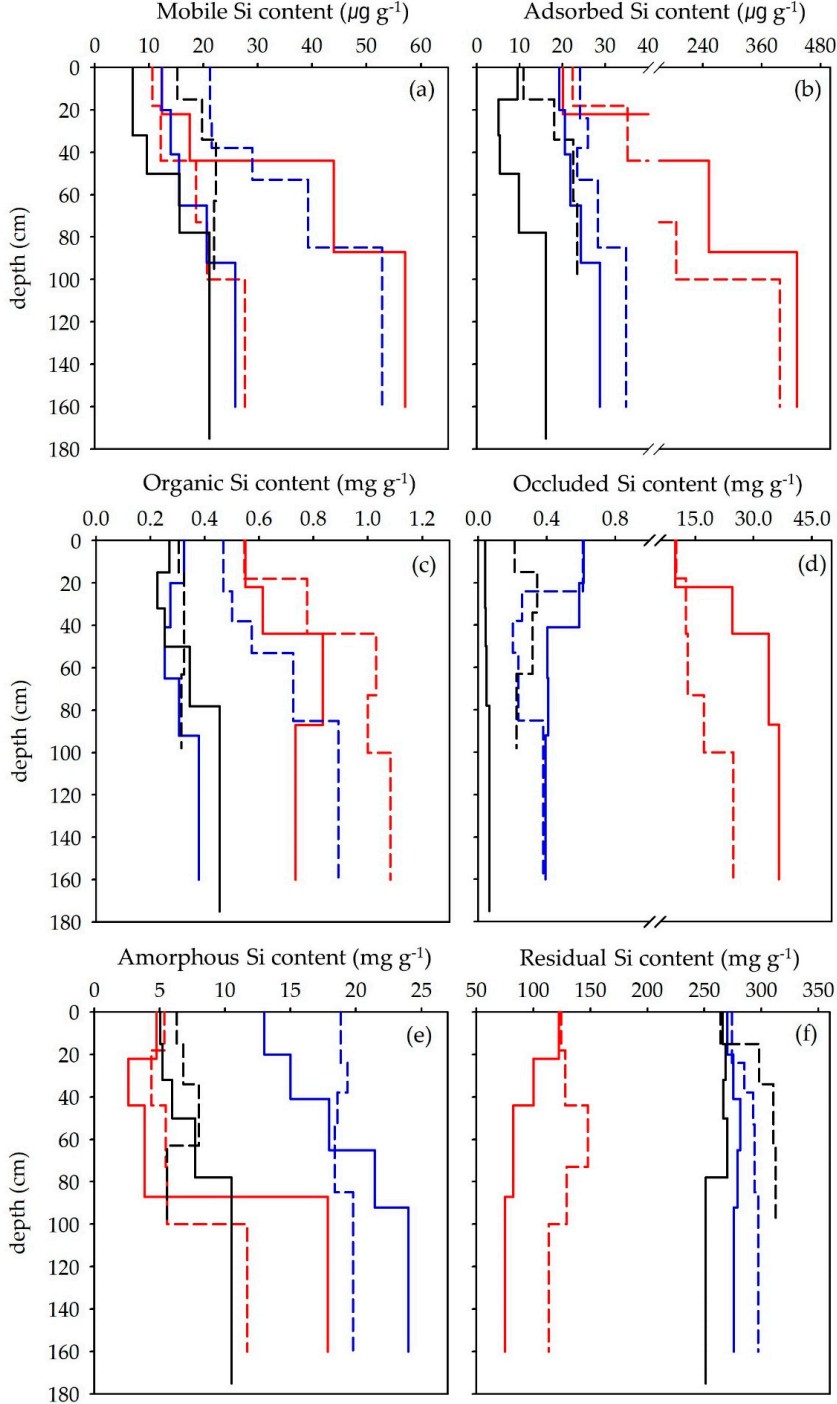

**Figure 3.** Depth fractionation of silicon in Jeju Island [Andisols; Noro (solid red), Pyeongdae (dashed red), Ultisols; Jeju (solid blue), and Alfisols; Gangjeong (dashed blue)] and Korean mainland forest soils [Inceptisols; Samgag (solid black) and Oesan (dashed black)] according to sequential extraction. (**a**) mobile Si, (**b**) adsorbed Si, (**c**) Si bound to soil organic matter, (**d**) Si occluded in pedogenic oxides and hydroxides, (**e**) Si in total amorphous silica, and (**f**) residual Si in crystalline silicates.

Si is chemically adsorbed by soil compounds such as oxides and hydroxides from Al and Fe, crystalline Fe oxides, allophane, and clay minerals [9,16,17,41]. In non-Andisols (Inceptisols from mainland Korea and Ultisols and Alfisols from Jeju Island), the mobile and adsorbed Si and the clay content had a positive correlation ($r$ = 0.682 and 0.858, respectively, $p < 0.01$ for each) (Table 4). Supporting findings by Georgiadis et al. [25], the Samgag series, which had the lowest clay content versus high sand content, showed the lowest mobile and adsorbed Si contents. In Andisols, the allophane and noncrystalline hydrous oxides of Fe and Al ($Al_o$ + 1/2$Fe_o$) contents increased with increasing soil depth (Table 2), and abundant crystalline Fe oxides as hematite and goethite were found (Table S1). These soil components had a positive correlation with mobile Si and adsorbed Si ($p < 0.01$, Table 4). Hence, this suggests that most of the DSi has been absorbed into subsoil horizons and moved downward during weathering.

**Table 4.** Spearman's rank correlation between readily soluble Si fraction (mobile Si and adsorbed Si) and selected physicochemical properties of the soils.

|  | Total ($n$ = 28) | | Non-Andisols ($n$ = 19) | | Andisols ($n$ = 9) | |
| --- | --- | --- | --- | --- | --- | --- |
|  | Mobile Si | Adsorbed Si | Mobile Si | Adsorbed Si | Mobile Si | Adsorbed Si |
| pH | 0.418 * | 0.756 ** | 0.608 ** | 0.505 * | 0.850 ** | 0.950 ** |
| Organic C | −0.444 * | 0.125 | −0.550 ** | −0.318 | −0.900 | −0.983 ** |
| Sand | −0.337 | −0.317 | −0.613 ** | −0.800 ** | 0.517 | 0.617 |
| Silt | 0.103 | 0.451 * | 0.270 | 0.547 * | −0.167 | −0.183 |
| Clay | 0.272 | 0.129 | 0.682 ** | 0.858 ** | −0.600 | −0.617 |
| $Fe_d$ | 0.161 | 0.711 ** | 0.544 * | 0.779 ** | −0.567 | −0.333 |
| $Al_o$ + 1/2$Fe_o$ | 0.164 | 0.720 ** | 0.057 | 0.426 | 0.883 ** | 0.883 ** |
| Allophane | - | - | - | - | 0.767 * | 0.767 * |
| $Fe_o/Fe_d$ | 0.174 | 0.691 ** | 0.080 | 0.439 | 0.750 * | 0.880 ** |

*,**: significant difference at 0.05 and 0.01 probability levels, respectively.

### 3.2.2. Si in Soil Organic Matter (SOM)

The SOM-bound Si content ranged from 0.2–1.0 mg·g$^{-1}$ across the profiles (Figure 3c). The A horizon in Andisols with higher organic carbon content showed 0.55 mg·g$^{-1}$, which was higher than the A horizon in other soils. While the organic carbon content in the soils decreased with increasing soil depth (Table 3), the Si content in the fractions rose with increasing soil depth or was similar to that in the A horizon. In particular, the Si content in the Jeju Island's Noro series and Pyeongdae series Andisols and the Gangjeong series Alfisols tended to increase noticeably with increasing soil depth. This may be because Si from amorphous and poorly crystalline silica, which are extracted from the latter fractions, was extricated by $H_2O_2$ treatment during the sequential extraction procedure [25].

Georgiadis et al. [32] reported that the Si content in SOM may be overestimated, since a large amount of Si is released from clay minerals, sesquioxides, and amorphous silica when treated by hot $H_2O_2$ solution. In this study, the Si content in SOM and the oxalate-extractable Si content from sequential extraction step 4 was correlated ($r$ = 0.651 **, Table S3). Furthermore, the Si content in SOM from soils other than Andisols was found to correlate with amorphous silica and clay fractions ($r$ = 0.603 ** and $r$ = 0.606 **, respectively) from sequential extraction step 5. As subsoil horizons from Jeju Island's soils have more amorphous and poorly crystalline silica, it is possible that the SOM-bound Si content was higher, and Si in these fractions can serve as a potential source for readily soluble Si.

### 3.2.3. Si Occluded in Pedogenic Oxides and Hydroxides

In Andisols, the content of Si occluded in pedogenic oxides and hydroxides increased with increasing soil depth, ranging from 10–36 mg·g$^{-1}$. In other soils, it ranged from 0.04–0.6 mg·g$^{-1}$, and the lowest content was found in the Samgag series (0.04–0.06 mg·g$^{-1}$) from mainland Korea (Figure 3d).

Acid ammonium oxalate reagents are known to be very effective in extracting short-range-order minerals with weak crystallinity, such as allophane, imogolite, and ferrihydrite [19,31,34]. Andisols in this study contained allophane in the range of 7.27–39.0% (Table 3). In Andisols, the content of Si occluded in pedogenic oxides and hydroxides showed a significant correlation with the allophane content ($r = 0.917$ **), which suggests that oxalate-extractable Si in these fractions constitutes allophane. Furthermore, the SEM/EDS analysis results of the Bw horizon from the Noro series Andisols before sequential extraction confirmed that allophane was exfoliated and formed on the surface of feldspars, and it was removed after acid ammonium oxalate extraction (step 4) (Figure 4).

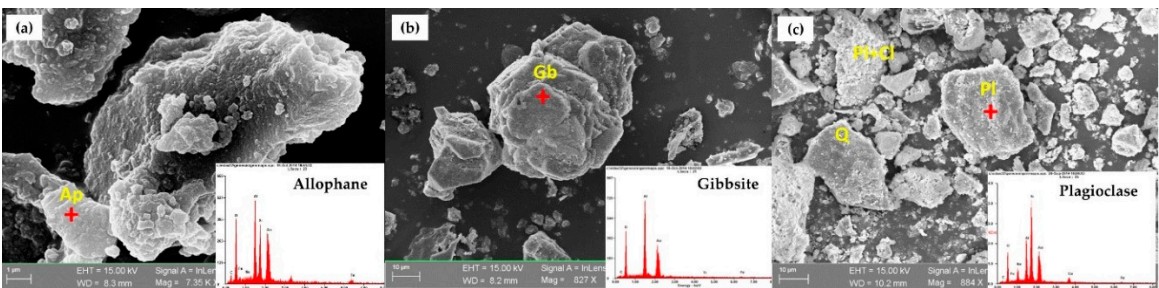

**Figure 4.** Scanning electron microscope/energy-dispersive X-ray spectroscopy (SEM/EDS) analysis after sequential extraction from Bw horizon of Noro in Jeju Island, showing (**a**) allophane formed on the surface of plagioclase before extracting, (**b**) removal of allophane after step 4, and (**c**) crystalline minerals after step 5. Ap: allophane, Gb: gibbsite, Q: Quartz, Pl: Plagioclase, Cl: Clay minerals.

Assuming that soil is lacking allophane if the acid ammonium oxalate extractable Si lower than 6 mg·g$^{-1}$ [42,43], a silicon phase only exists in Andisols. In such soils, oxalate-extractable Si and oxalate-extractable Al ($r = 0.555$ *) and Fe ($r = 0.594$ **) were correlated. This is the result of Si extracted from the soils, which was occluded or co-precipitated with noncrystalline hydrous oxides of Fe and Al [25,44].

### 3.2.4. Si in Amorphous Silica

In soil Si pools, amorphous silica includes pedogenic and biogenic silica [32]. In amorphous silica, the Si content was the highest in Jeju Island's Ultisols and Alfisols, ranging from 13.0–24.0 mg g$^{-1}$. In contrast, Andisols had the lowest Si content with a range of 2.62–5.58 mg g$^{-1}$ from the A horizon to the Bw horizon. However, the Si contents in the C horizon in the Noro series and the BC horizon in the Pyeongdae series were 17.8 and 11.7 mg g$^{-1}$, respectively (Figure 3e).

Amorphous silica is formed by precipitation in the soil solution where silica is over-saturated due to surface evaporation [9,19]. Amorphous silica accumulation may be related to repeated redox changes if Si bound to Fe oxides is present [45,46]. In soils other than Andisols, the Si content in amorphous silica had a close correlation with clay fractions ($r = 0.93$ **). This demonstrates the accumulation of amorphous silica in clayey horizons [25,45]. Drouza et al. [47] reported that, in the soils in Nisyros Island, Greece, where volcanic materials are developed, the SiO$_2$-rich parent material combined with low levels of rainfall restricted leaching, and an extended dry season resulted in the accumulation of Si in the soil solution and its deposition in amorphous forms.

As the two soils that develop into Ultisols and Alfisols on Jeju Island have pyroclastic materials as their parent material, weathering takes place at a rapid pace. However, these soils develop in areas where precipitation is relatively low and evapotranspiration is high rather than in areas with Andisols [23,48], which increases the Si concentration in the soil solution, inhibiting the pedogenic process of allophane and producing amorphous silica in large quantities. The Yongdang series developed in the western and norther regions of Jeju Island where non-Andisols, such as the two tested soils in this study, contain fragipan in the subsoil horizons [49]. The formation of fragipan in non-Andisols is associated with Si accumulation [9].

After NaOH extraction, XRD analysis for the Bw horizon from the Noro series Andisols showed that it is composed of primary minerals, such as plagioclase, orthoclase, pyroxene, olivine, amphibole, and magnetite; and secondary minerals, such as clay minerals, iron oxides, and quartz (Table 5, Figure S2). In addition, SEM/EDS analysis found that gibbsite was removed by NaOH extraction, while crystalline minerals remained (Figure 4). As with the results of Georgiadis et al. [25], 0.2M NaOH solution at 20–25 °C did not extract crystalline silicate in Andisols in this study while amorphous and poorly crystalline clay compounds seem to have been fully extracted.

**Table 5.** X-ray diffraction (XRD) semi-quantitative analysis after sequential extraction from the Bw horizon of Noro.

| Mineral Compositions | Untreated | NH$_4$-Oxalate | NaOH |
|---|---|---|---|
| | **wt.%** | | |
| Quartz | 10.9 | 12.7 | 14.9 |
| Plagioclase | 16.6 | 10.5 | 24.9 |
| K-Feldspars | 4.0 | 4.2 | 4.2 |
| Micas/Illite | 5.4 | 5.1 | 3.3 |
| Hornblende | 2.0 | 3.1 | 3.2 |
| Pyroxene | 5.7 | 7.3 | 15.6 |
| Olivine | 9.5 | 9.5 | 7.7 |
| Kaolinite | 0.7 | 0.0 | 0.0 |
| Chlorite | 3.0 | 3.7 | 4.0 |
| Gibbsite | 19.5 | 26.6 | 0.0 |
| Hematite/Goethite | 16.8 | 12.7 | 17.5 |
| Magnetite | 5.9 | 4.5 | 4.8 |

### 3.2.5. Residual Si

The Si of the residual fraction comprises Si in primary and secondary minerals. The residual Si content in Ultisols and Alfisols from Jeju Island and Inceptisols from mainland Korea ranged from 251–312 mg·g$^{-1}$, which was two times higher than that in Jeju Island's Andisols. The residual Si in Andisols decreased with an increasing soil depth, and the Si of the lowest residual fraction was 74.8 mg·g$^{-1}$ in the C horizon from the Noro series (Figure 3f).

Not only amorphous silica but also a large amount of quartz and layer silicate minerals such as kaolinite, illite, chlorite, and vermiculite are produced in Ultisols and Alfisols from Jeju Island. As plagioclase and feldspars remained in Inceptisols from mainland Korea, the residual Si content was high (Table S1) [50]. In contrast, since Al formed Al-humus complexes in surface horizons from Jeju Island's Andisols during pedogenesis of volcanic ash soils, DSi moved downward. Since Al-rich allophane was mostly dissolved, as the Si concentration in the soil solution was low in subsoil horizons, DSi continued to be leached, leading to a very low residual Si content.

### 3.3. Relationships between Readily Soluble Si and Other Si Fractions in Soils

Table 6 describes the correlation between mobile and adsorbed Si and other Si fractions. In the study soils, the mobile Si and adsorbed Si contents had a close correlation ($r = 0.661$, $p < 0.01$). In subsoil horizons from the Samgag and Gangjeong series, the adsorbed Si content was lower than the mobile Si content. The Bw and C horizons from the Noro series Andisols showed a higher mobile Si content than did the other soils. These results are attributable to the fact that, since there is no boundary between mobile and loosely adsorbed Si in the soils, some loosely adsorbed Si was already desorbed from the soils at sequential extraction step 1 with CaCl$_2$ [25]. Consequently, the sum of the two fractions is a readily soluble, easy-to-mobilize Si fraction in Si pools from the soils [25,51]. As mentioned earlier, Jeju Island's Andisols, which have a high content of soil components that absorb monosilicic acid, have a larger amount of readily soluble Si in subsoil horizons compared with mainland Korea's Inceptisols.

**Table 6.** Spearman's rank correlation between readily soluble Si fraction (mobile Si and adsorbed Si) and potential sources in Si fraction ($Si_{org}$, $Si_{occ}$, $Si_{ma}$).

| | Total (*n* = 28) | | Non-Andisols (*n* = 19) | | Andisols (*n* = 9) | |
|---|---|---|---|---|---|---|
| | Mobile Si | Adsorbed Si | Mobile Si | Adsorbed Si | Mobile Si | Adsorbed Si |
| Adsorbed Si | 0.661 ** | | 0.849 ** | | 0.933 ** | |
| Organic Si | 0.539 ** | 0.814 ** | 0.819 ** | 0.672 ** | 0.536 | 0.644 |
| Occluded Si | 0.256 | 0.782 ** | 0.251 | 0.591 ** | 0.900 ** | 0.933 ** |
| Amorphous Si | 0.514 ** | 0.116 | 0.643 ** | 0.839 ** | 0.450 | 0.550 |
| Residual Si | 0.216 | −0.286 | 0.691 ** | 0.655 ** | −0.533 | −0.450 |

*,**: significant difference at 0.05 and 0.01 probability levels, respectively.

In the study soils, adsorbed Si had a correlation with SOM-bound Si ($r$ = 0.814 **) and occluded Si ($r$ = 0.782 **), while mobile Si had a correlation with Si in amorphous silica ($r$ = 0.514 **) (Table 6). In Si pools of the soils, Si in allophane and amorphous silica fractions is more soluble than crystalline minerals, so not only is Si dissolved more easily, but the desorption of monosilicic acid adsorbed to mineral surfaces may also become a potential source for readily soluble Si (mobile + adsorbed Si) [17,25]. The solubility of amorphous Si ranges between 1.8–2 mM compared with quartz's 0.10–0.25 mM Si [17,52] Miretzky et al. [27] reported that, under a wet climate in Buenos Aires Province, Argentina, amorphous Si of volcanic pyroclastic or biogenic origin is likely to determine a high concentration of DSi in groundwater.

Especially in Andisols, mobile and adsorbed Si had the highest correlation with occluded Si ($r$ = 0.900 ** and $r$ = 0.933 **, respectively). Occluded Si comprises allophane, and allophane with high adsorption capacity for monosilicic acid can make the biggest contributions to readily soluble Si [41,53]. During weathering under high precipitation conditions, allophane produces gibbsite by extreme desilication [19,21,22]. Subsoil horizons from Andisols with high allophane content contain more than 17% gibbsite content (Table S1, Figure 4). The formation of gibbsite in Andisols means that a large amount of Si has been leached and has moved down the soil, which may affect the Si content in groundwater.

Among extractants attempted by Georgiadis et al. [32], 0.01 M acetic acid showed considerable desorption efficiency with the smallest effect on clay minerals (except for smectite) and silica gel. Since the soils did not include smectite, adsorbed Si may not have been released from clay minerals. This concentration of acetic acid may have had an effect on allophane, a short-range-order mineral produced from Andisols in the study soils. However, since 0.01 M acetic acid had the smallest amount of Si released from the test materials compared with that of other extractants [32], Si released from allophane in this study's soils may also have been minimized.

*3.4. Percentaget Distribution of the Different Si Fraction to Total Si*

The distribution percentage of the total Si content in the soils by form is provided in Table 7. The largest amount of Si in the soils was residual Si in primary and secondary silicates, accounting for 57–98% of the total Si. Residual Si showed the largest distribution across the profiles of the Samgag and Oesan series derived from mainland Korea's granite and granite gneiss Inceptisols, while Si extracted from amorphous silica (including pedogenic and biogenic silica) is derived from basalt and has pyroclastic materials as its parent material. The Jeju and Gangjeong series on Jeju Island, where Ultisols and Alfisols are developed, and Inceptisols from mainland Korea had the second-largest amount of Si (1.74–8% of total Si). Meanwhile, Si occluded in pedogenic oxides and hydroxides showed the second-highest percentage (7.2–28% of total Si) in Jeju Island's Andisols. For SOM, Si was around 0.5% in Andisols, which was five times higher than that in other soils. Mobile Si and adsorbed Si showed the smallest distribution across the soil profiles. Mobile Si's distribution was 0.003–0.04% of the total Si. In soils other than Andisols, adsorbed Si's distribution was similar to that of mobile Si.

However, adsorbed Si in the C horizon from the Noro series Andisols and the BC horizon from the Pyeongdae series was approximately 0.3% of the total Si, which was 10 times higher than that in other soils. Furthermore, mobile Si was the highest in the Bw and C horizons from the Noro series at about 0.04% of the total Si.

**Table 7.** Distribution of different Si fractions by sequential extraction.

| Horizon | Depth | Mobile Si | Adsorbed Si | Organic Si | Occluded Si | Amorphous Si | Residual Si |
|---|---|---|---|---|---|---|---|
| | (cm) | % | | | | | |
| Noro | | | | | | | |
| A | 0–22 | 0.009 | 0.015 | 0.40 | 7.19 | 3.46 | 88.9 |
| BA | 22–44 | 0.014 | 0.037 | 0.48 | 19.2 | 2.05 | 78.2 |
| Bw | 44–87 | 0.036 | 0.209 | 0.69 | 28.0 | 3.19 | 67.9 |
| C | 87–160 | 0.044 | 0.332 | 0.56 | 27.9 | 13.7 | 57.5 |
| Pyeongdae | | | | | | | |
| A | 0–18 | 0.008 | 0.016 | 0.39 | 7.16 | 3.82 | 88.6 |
| AB | 18–44 | 0.008 | 0.024 | 0.53 | 8.61 | 3.00 | 87.8 |
| Bw1 | 44–73 | 0.011 | 0.077 | 0.62 | 7.80 | 3.27 | 88.2 |
| Bw2 | 73–100 | 0.013 | 0.121 | 0.65 | 11.2 | 3.65 | 84.3 |
| BC | 100–160 | 0.018 | 0.262 | 0.72 | 16.3 | 7.72 | 74.9 |
| Jeju | | | | | | | |
| Ap | 0–20 | 0.004 | 0.007 | 0.11 | 0.22 | 4.57 | 95.1 |
| AB | 20–41 | 0.005 | 0.007 | 0.09 | 0.20 | 5.15 | 94.5 |
| Bt1 | 41–65 | 0.005 | 0.007 | 0.08 | 0.13 | 5.98 | 93.8 |
| Bt2 | 65–92 | 0.007 | 0.008 | 0.10 | 0.14 | 7.12 | 92.6 |
| Bt3 | 92–160 | 0.009 | 0.010 | 0.13 | 0.13 | 7.99 | 91.7 |
| Gangjeong | | | | | | | |
| Ap | 0–24 | 0.007 | 0.008 | 0.16 | 0.21 | 6.41 | 93.2 |
| BAt | 24–38 | 0.007 | 0.008 | 0.16 | 0.08 | 6.34 | 93.4 |
| Bt1 | 38–53 | 0.009 | 0.007 | 0.18 | 0.06 | 5.95 | 93.8 |
| Bt2 | 53–85 | 0.013 | 0.009 | 0.23 | 0.07 | 5.87 | 93.8 |
| Bt3 | 85–160 | 0.017 | 0.011 | 0.28 | 0.12 | 6.22 | 93.4 |
| Samgag | | | | | | | |
| A | 0–15 | 0.003 | 0.003 | 0.10 | 0.01 | 1.85 | 98.0 |
| BA | 15–32 | 0.003 | 0.002 | 0.08 | 0.01 | 1.90 | 98.0 |
| Bw | 32–50 | 0.004 | 0.002 | 0.09 | 0.02 | 2.18 | 97.7 |
| C1 | 50–78 | 0.006 | 0.003 | 0.12 | 0.02 | 2.77 | 97.1 |
| C2 | 78–180 | 0.008 | 0.006 | 0.17 | 0.02 | 4.00 | 95.8 |
| Oesan | | | | | | | |
| A | 0–15 | 0.006 | 0.004 | 0.11 | 0.08 | 2.33 | 97.5 |
| BA | 15–34 | 0.006 | 0.006 | 0.11 | 0.11 | 2.22 | 97.5 |
| Bw | 34–63 | 0.007 | 0.007 | 0.10 | 0.10 | 2.51 | 97.3 |
| C | 63–98 | 0.007 | 0.007 | 0.10 | 0.07 | 1.74 | 98.1 |

The weathering of Jeju Island's Andisols is fast because the parent material is composed of pyroclastic deposits [23]. Under conditions where many organic matter sources exist due to high precipitation, Al preferentially forms humus complexes, which may lead to excessive Si [18–20]. However, since many silicic acids are leached from surface horizons to subsurface horizons during heavy rainfall, the readily soluble Si content is relatively low in the soil profile's upper part. In subsoil horizons, Si released by the weathering of primary minerals and soluble Si leached from the upper profile combines with polymerized Al to form allophane [19,20]. As the island has more than 2000 mm precipitation annually on average, significant leaching of Si takes place, which leads to a lower soluble Si concentration in the soil solution and the formation of Al-rich allophane with a lower Si content [24,48]. During Andisol formation, noncrystalline materials (e.g., allophane, ferrihydrite, and noncrystalline hydrous oxides of Fe and Al) are formed more abundantly than crystalline layer silicate. Noncrystalline

materials present in Andisols can play an important role in adsorption of Si released during weathering, which may increase the readily soluble Si content. Therefore, in the volcanic ash soil of Jeju Island, where precipitation is high and water permeability is fast, the readily soluble Si fraction from the soil Si pools is released as DSi in the soil solution. This is likely to move into groundwater by infiltration. Non-Andisols in Jeju Island and the Korean mainland mainly produce quartz and layered silicates. As a result, Si is not leached much in non-Andisols.

## 4. Conclusions

This study was conducted to investigate the Si distribution and content in Andisols, Ultisols, and Alfisols in Jeju Island, and Inceptisols derived from granite and granite gneiss in the Korean mainland. The Si occurring as different soil compounds can be soluble, and could eventually migrate underground to increase the $SiO_2$ concentration. The average $SiO_2$ concentration in Jeju Island's groundwater was about four times higher than that in mainland Korea's groundwater. According to the results of Si sequential extraction, more than 90% of Jeju Island's Ultisols and Alfisols and mainland Korea's Inceptisols existed as crystalline Si and showed no difference by soil depth. While Jeju Island's Andisol soil had a smaller distribution percentage of crystalline Si compared with the non-Andisols of Jeju Island and mainland Korea, the percentage of Si constituting allophane, which is a noncrystalline material, absorbed Si. Soluble Si was noticeably higher. In the topsoil of Jeju Island's Andisols, Al-humus complexes are mostly formed, while allophane with a lower Si content is produced in the subsoil, which suggests that DSi is leached by high precipitation during weathering. At the same time, soluble or potentially soluble Si, a large amount of which is contained during the pedogenesis of Andisols, continues to be leached. This may be a factor in increasing the $SiO_2$ content in groundwater, which helps to improve the taste of drinking water and benefits human health.

**Supplementary Materials:** The following are available online at http://www.mdpi.com/2073-4441/12/10/2686/s1: Table S1: Mineral compositions of selected soil samples, Table S2: Results of the sequential silicon, aluminum, and iron extraction of analyzed soils, Table S3: Spearman's rank correlation between different Si fractions of analyzed soils, Figure S1: Silicon released by extraction with 0.2 M NaOH at a soil:solution ratio of 1:400 at 20–25 °C, Figure S2: X-ray diffraction pattern after sequential extraction from the Bw horizon of the Noro series.

**Author Contributions:** Conceptualization and methodology, W.-P.P. and H.-N.H.; formal analysis, W.-P.P.; data curation, W.-P.P. and H.-N.H.; writing—original draft, W.-P.P.; writing—review & editing, H.-N.H. and B.-J.K. All authors have read and agreed to the published version of the manuscript.

**Funding:** This research received no external funding.

**Acknowledgments:** We would like to express our gratitude to Seong-Taek Yun at Korea University who kindly provided $SiO_2$ concentration data in groundwater from the Korean mainland and also give thanks to Hyomin Lee for his technical assistance.

**Conflicts of Interest:** The authors declare no conflict of interest.

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
