# Peer review of "Silicon Fractionation of Soluble Silicon in Volcanic Ash Soils That May Affect Groundwater Silicon Content on Jeju Island, Korea"

_water, doi:10.3390/w12102686_

Round 1
Reviewer 1 Report
Interesting paper focused on quantification of the different Si fractions in volcanic ash soils on that may affect groundwater Si content, , using the sequential extraction method.
Aims clearly stated. Good introduction, clear, comprehensive, well supported by numerous and updated references.
Good presentation of materials.
In general, adequate description of methodology, supported by references.
Results presentation is clear, with a nice set of figures.
Discussion is, in general, well done, supported by data and references.
Conclusions are coherent and logical.
Just 1 suggestion and 1 question:
- A figure with study area and sampling points location.
- On line 311, “clay” means clay fraction or clay minerals?
Author Response
Attached is the response to Reviewer 1 comments

Reviewer 2 Report
The paper done by Won-Pyo Park and Co-As shows a very nice work in regard to the silica occurrence in soils of South Korea. Although the paper is well arranged and structured, the written English needs some improvement as sometimes the text reading and comprehension is difficult. The scientific results are very interesting and provide an excellent example of how the silica occurrence in continental waters should be approached as they carefully consider any silica reservoir in the area. The paper deserves publication after english improvement as there are several issues with grammar and English language expression. Furthermore, I am missing some maps where the study area at regional scale, and at a higher scale of materials, and outcrops (e.g., volcanic, granitic materials, as well as soils), and sampling locations should be shown. A geological and/or pedological map would be very useful indeed as a Figure 1. In the following, I list some examples of how the English should be clearly improved, though there are other part of the text where it should also be corrected:
Lines 68 to 71: I would split and rewrite the sentence: “Georgiadis et al. [25] reported that water-soluble and adsorbed Si, among other Si fractions from Si sequential extraction, is easy to mobilize, and amorphous Si and Si occluded in pedogenic oxides and hydroxides can become potential sources of easy-to-mobilize Si because amorphous Si is dissolved more easily than crystalline minerals [26]” to “Georgiadis et al. [25] reported that water-soluble and adsorbed Si, among other Si fractions from Si sequential extraction, is easy to mobilize. In addition, amorphous Si and Si occluded in pedogenic oxides and hydroxides can become potential sources of easy-to-mobilize Si because it is dissolved faster than crystalline phases [26]”
Lines 90 to 91: Please rework the phrase “Concentrated nitric acid was put into the samples to reach a 0.5% nitric acid concentration.” to “A solution of concentrated nitric acid at X % was put into the samples to reach a 0.5% nitric acid concentration.” Indeed, it is not described how concentrated as the nitric acid solution. Should also be mentioned.
Line 93: replace “designated” by “calculated”
Lines 188 to 189: Please rework the three lines: “Furthermore, while amorphous silica can be extracted as biogenic and minerogenic Si [32], both these forms are potential sources for mobile Si [25].” I would suggest something like “As amorphous silica can be extracted as biogenic and minerogenic Si [32], both reservoirs are potential sources for mobile Si [25]”
Line 281: Please replace “ingredients” by “compounds”
Lines 285 to 286: Rework the phrase “which had the lowest clay content and high sand content”. I suggest “which had the lowest clay versus high sand contents”
Lines 290 to 291: Rework the lines “it is believed that, as dissolved Si moved downward during weathering, much of it may have been absorbed into subsoil horizons.” To “This suggests that most of the dissolved Si has been absorbed into subsoil horizons as moved downward during weathering.”
Lines 329 to 330: Rework the lines “Since soil is not considered to contain allophane if the acid ammonium oxalate extractable Si value is 6 mg·g−1 or less [42, 43], allophane did not exist in soils other than Andisols.” I suggest to “Assuming that soil is lacking allophane if the acid ammonium oxalate extractable Si lower than 6 mg·g−1 [42, 43], such a silicon phase only exists in Andisols.”
Lines 338 to 342: Please rework “By contrast, Andisols showed the lowest range of 2.62 to 5.58 mg·g-1 from the A horizon to the Bw horizon. However, the C horizon from the Noro series and the BC horizon from the Pyeongdae series were 17.8 340 and 11.7 mg·g-1, respectively (Figure 2e).”
Lines 354 to 356: Please rework “In Jeju Island’s west and north where non-Andisols such as these two study soils are distributed, the Yongdang series has developed [49], containing frangipans, which are correlated to the accumulation of Si [9].”
Lines 357 to 360: Please rework “XRD analysis results for the Bw horizon from the Noro series Andisols showed that its primary minerals, such as plagioclase, orthoclase, pyroxene, olivine, amphibole, and magnetite, and its secondary minerals, such as clay minerals, iron oxides, and quartz, remained after NaOH extraction (Table 5).” I suggest to “After NaOH extraction, XRD analysis for the Bw horizon from the Noro series Andisols showed that is composed of primary minerals, such as plagioclase, orthoclase, pyroxene, olivine, amphibole, and magnetite; and secondary minerals, such as clay minerals, iron oxides, and quartz (Table 5).”
Lines 361 to 363: Please rework “Supporting findings by Georgiadis et al. [25], 0.2M NaOH extraction at 20–25°C appears not to have attacked crystalline silicate in Andisols, and amorphous and poorly crystalline clay compounds seem to have been fully extracted.”
Lines 457 to 458: Please rework “As Jeju Island’s Andisols have pyroclastic materials as their parent material, their weathering is fast [23].” I sugges to “The weathering of Jeju Island’s Andisols as the parent material is composed of pyroclastic deposits [23].”
Lines 465 to 469: Please rework “As a result, since there are many noncrystalline materials (e.g., allophane, ferrihydrite, and noncrystalline hydrous oxides of Fe and Al) in subsoil horizons, we believe that Si released during weathering is absorbed, which may increase the readily soluble Si content. Meanwhile, as non-Andisols from Jeju Island and mainland Korea mainly produce quartz and layered silicates, Si is not leached very much.” as there are several grammar problems.
Lines 471 to 473: Please rework “This study was conducted to investigate the distribution characteristics of Si, which exists in various forms in soils, and to estimate the Si content, which is soluble or potentially soluble and moves to groundwater to increase the SiO2 concentration” which could be “This study was conducted to investigate the silica distribution and content in “XXX region/materials/outcrops”. The Si occurring as different soil compounds can be soluble, and could eventually migrate underground to increase the SiO2 concentration”.
Author Response
Attached is the response to Reviewer 2 comments
